# A Review on Bioactive Anthraquinone and Derivatives as the Regulators for ROS

**DOI:** 10.3390/molecules28248139

**Published:** 2023-12-17

**Authors:** Lihua Zhao, Lin Zheng

**Affiliations:** 1Tianjin Renai College, Tianjin 301636, China; huazi.95@163.com; 2College of Pharmaceutical Engineering of Traditional Chinese Medicine, Tianjin University of Traditional Chinese Medicine, Tianjin 301617, China

**Keywords:** natural anthraquinone, anthraquinone derivatives, ROS, structure-activity relationship, antioxidant mechanism

## Abstract

Anthraquinones are bioactive natural products, which are often found in medicinal herbs. These compounds exert antioxidant-related pharmacological actions including neuroprotective effects, anti-inflammation, anticancer, hepatoprotective effects and anti-aging, etc. Considering the benefits from their pharmacological use, recently, there was an upsurge in the development and utilization of anthraquinones as reactive oxygen species (ROS) regulators. In this review, a deep discussion was carried out on their antioxidant activities and the structure-activity relationships. The antioxidant mechanisms and the chemistry behind the antioxidant activities of both natural and synthesized compounds were furtherly explored and demonstrated. Due to the specific chemical activity of ROS, antioxidants are essential for human health. Therefore, the development of reagents that regulate the imbalance between ROS formation and elimination should be more extensive and rational, and the exploration of antioxidant mechanisms of anthraquinones may provide new therapeutic tools and ideas for various diseases mediated by ROS.

## 1. Introduction

According to the biological theory of free radicals, the generation and scavenging of free radicals should be in a dynamic balance under normal circumstances. The broken balance of free radicals will harm the body and cause diseases. Reactive oxygen species (ROS), which include several kinds of oxygen related free radicals, are of Janus-faced nature in cells. On one hand, ROS signaling is of vital importance in maintaining crucial cellular functions. They are involved in the signaling cascades and the regulation of cellular processes. On the other hand, they provide detrimental damages to macromolecules. The overproduction of ROS can damage cells and induce a variety of diseases, such as neurodegenerative diseases, cancer, hepatopathic diseases, and cardiovascular diseases, etc. [1]. Mechanisms of ROS mediated diseases have also been studied. For example, the brains often show oxidative damage in neurodegenerative diseases. Alzheimer’s disease (AD) is a classic example. Long-lasting ROS generation via amyloid beta (Aβ)-Cu(II) peptides and mitochondrial dysfunction may lead to neuronal damage and ultimately dementia. Elevated ROS can damage DNA, transform normal cells into cancer cells, trigger enhanced epidermal growth factor receptor signaling and promote tumor progression [2,3,4]. Excessive ROS can lead to fibrosis, liver inflammation and hepatocellular carcinom [5]. Endothelial cells and vascular smooth muscle cells, the most important cells for maintaining the integrity and homeostasis of the vascular system, both represent targets of ROS and ROS signaling. Excessive ROS may damage vascular cells, induce the proliferation and migration of vascular smooth muscle cells, and then cause vascular remodeling [6,7]. Therefore, therapeutic strategy targeting ROS and ROS-generating system is to maintain the dynamic balance of ROS.

Antioxidants are a well-studied class of reagents that mainly include enzymes and non-enzymes. Enzymatic antioxidants, including superoxide dismutase (SOD), catalase (CAT), glutathione peroxidase (GPX), glutathione reductase (GR), and glutathione S-transferases (GST), are involved in ROS production and peroxides reduction, thus playing a major role in maintaining the redox balance. Nonenzymatic antioxidants, such as polyphenols, anthocyanins, polyenes, etc., can directly increase rates of ROS degradation and halt excessive ROS accumulation [8,9]. Another class of reagents can interact indirectly with signaling pathways or key molecules involved in the production and elimination of ROS to reduce oxidative damage [10].

Natural nonenzymatic antioxidants have attracted the attention of researchers because of their wide distribution and easy availability [11,12]. The largest group of natural quinones are phenol-substituted anthraquinones [13], which are abundant in many plants, such as *Rheum rhabarbarum* L. (Polygonaceae), *Polygonum cuspidatum* Sieb. et Zucc. (Polygonaceae), *Polygonum multiflorum* Thunb. (Polygonaceae), *Rubia cordifolia* Thunb. (Rubiaceae), *Cassia sieberiana* DC. (Caesalpiniaceae), *Rhamnus dahurica* Pall. (Rhamnaceae), *Salvia mitiorrhiza* Bge. (Labiatae), etc. [14,15,16,17,18,19]. The anthraquinone parent structure has a three fused-ringed structure with two carbonyl groups (Figure 1). In the structure of anthraquinone derivatives, the anthraquinone ring is the basic scaffold, with the substituent groups connected to the position of C_1_–C_8_ [13,20]. Naturally occurring anthraquinones are often substituted with hydroxyl, hydroxymethyl, methyl, methoxy, and carboxyl groups on the parent structure and are present in the plant either in free form or in combination with sugars to form glycosides. The structures of some natural anthraquinone compounds are shown in Figure 1. Recently, a large number of studies have shown that natural anthraquinone compounds have not only strong antioxidant activity but also little toxicity or few side effects [21,22,23,24]. Therefore, the natural anthraquinone compounds are particularly attractive and have great potential as antioxidant agents. In this work, the antioxidant potential of bioactive anthraquinone compounds has been reviewed. The main line of this work is showed in Figure 2.

## 2. Free Radicals and Oxidative Stress

Free radicals are present in any chemical entity with one or more unpaired electrons in the outer orbital [25]. Humans are aerobic organisms, so the body inevitably produces free radicals. The sources of free radicals can be endogenous or exogenous [26]. The most important endogenous sources are mitochondrial respiratory chain enzymes, reduced nicotinamide adenine dinucleotide phosphate (NADPH) oxidase, nitric oxide synthase and xanthine oxidase. At the same time, external factors, such as environmental pollutants, radiation, smoking and drugs, may also contribute to the generation of free radicals [27,28,29,30].

ROS or reactive nitrogen species (RNS) are products of cell metabolism [31]. ROS can be in radical or nonradical forms, and they may interchange with each other in the body. Common free radical ROS include hydroxyl radicals (^•^OH), hydroperoxyl radicals (HO_2_^•^), superoxide anions (O_2_^•−^), nitric oxide (NO^•^) and peroxy radicals (ROO^•^) [32]. In addition, the common nonradical ROS are hydrogen peroxide (H_2_O_2_), singlet oxygen (^1^O_2_), hypochlorous acid (HOCl), and peroxynitrite (ONOO^−^) [33,34].

ROS have both beneficial and harmful effects on biological systems [31]. The delicate balance between these two effects is achieved through the mechanism of “redox regulation” [35]. Oxidative stress is caused by the imbalance between the antioxidant defense system and the intracellular accumulation of ROS [36]. The excessive production of ROS and oxidative stress play vital roles in several kinds of life-threatening diseases, such as Alzheimer’s disease, liver injury, and cancer. Consequently, it is significant to counteract the harmful effects generated by ROS, which is possible through the use of antioxidants [37].

Metals play an important role in the processes of ROS formation and elimination [38,39]. Metal ions with redox activity, such as iron and copper, are cofactors of various enzymes. However, the iron and copper ions may also produce ROS and further damage the organism in the process of reactions [40]. For example, Fe^2+^ is necessary for DNA synthesis, hemoglobin synthesis and electron transport [41]. However, Fe^2+^ could catalyze the Haber–Weiss reaction to generate more harmful ^•^OH from O_2_^•−^. When H_2_O_2_ reacts with Fe^2+^, the Fenton reaction can occur to form ^•^OH [42].
Fe^2+^ + H_2_O_2_ → ^•^OH + OH^−^ + Fe^3+^
Fe^3+^ + O_2_^•−^ → Fe^2+^ + O_2_
Or merged into: O_2_^•−^ + H_2_O_2_ → O_2_ + ^•^OH + OH^−^

Cu^2+^ catalyzes the oxidation of low-density lipoprotein, which has a similar Fenton reaction, directly causing ROS production and significantly reducing glutathione (GSH) levels with exposure to high concentrations of Cu^2+^ [43]. Zinc is also an essential nutrient for the maintenance of all life forms. Zn^2+^ is the most abundant metal in the brain, mainly protecting the mercapto group of proteins and enzymes from oxidation or inhibiting the formation of ^•^OH by H_2_O_2_ through the Fenton reaction [44]. ^•^OH is the most destructive species in free radical pathology and can induce oxidative damage in almost all cell molecules [45].

A detailed illustration of ROS formation is shown in Appendix A.

## 3. Theoretical Study on the Antioxidant Activity of Anthraquinone Compounds

### 3.1. General Mechanism of Antioxidant Activity of Anthraquinone Compounds with Phenolic Substituents

Living cells contain both low molecular weight antioxidants and high molecular weight antioxidant enzymes, both of which prevent and repair damage caused by free radicals. Antioxidants, according to their properties, can be divided into two types: preventive antioxidants [46] and chain-breaking antioxidants. Preventive antioxidants can scavenge free radicals in the chain initiation stage, such as SOD, CAT and metal (iron and copper) chelators. Chain-breaking antioxidants, often referred to as free radical scavengers, can capture peroxy radicals in the free radical reaction chain and prevent or slow down the progress of the free radical chain reaction. A large number of studies have shown that phenolic hydroxyl groups, such as *ortho*-phenolic hydroxyl groups and *para*-phenolic hydroxyl groups, could react with free radicals to produce stable semiquinone free radicals.

Phenolic compounds have three main antioxidant mechanisms that are widely accepted [19,47,48,49]. The first is the hydrogen atom transfer (HAT) mechanism, in which R^•^ free radicals attract a hydrogen atom from the antioxidant ArOH (Equation (1)) [50,51,52]. In this mechanism, the bond dissociation enthalpy (BDE) of the O-H bond is a significant parameter for scavenging activity evaluation. Although BDE is a thermodynamic parameter, the BDE of the same type of antioxidant correlates well with the logarithm of radical-scavenging rate constants (log*K*_s_) [53].
R^•^ + ArOH → RH + ArO^•^(1)

The second kind of mechanism is a single-electron transfer followed by proton transfer (SET-PT), namely, the rate determining step of ArOH losing one electron to generate ArOH^+•^ followed by a proton transfer step (Equation (2)) [54].
R^•^ + ArOH → R^−^ + ArOH^+•^ → RH +ArO^•^(2)

In the SET-PT mechanism, the ionization potential (IP) is the most important parameter for evaluating antioxidant activity. The lower the IP value is, the easier the formation of phenoxy radicals [50]. Nevertheless, the extremely low IP may convert the antioxidant to a prooxidant, because prooxidants can generate O_2_^•−^ by directly transferring an electron to the surrounding O_2_ [55].

The third antioxidant mechanism is sequential proton loss and electron transfer (SPLET). ArOH loses protons to form ArO^−^, and then electron transfer occurs [50,52,56,57]. According to Equation (3), the reaction enthalpy change is related mainly to the proton affinity potential of the anion ArO^−^, and the subsequent reaction enthalpy change is related mainly to the electron transfer enthalpy.
ArOH → ArO^−^ + H^+^(3)
ArO^−^ + R^•^ → ArO^•^ + R^−^
R^−^ + H^+^ → RH

The mechanism adopted is also related to the solvent. HAT is clarified to be the most favorable mechanism to describe the antioxidant activity of hydroxyanthraquinone in the gas phase and non-aqueous solvent (e.g., benzene). While in a aqueous solution, SPLET represents the most reasonable reaction pathway in thermodynamics. Because of the intermolecular hydrogen bond between water and phenoxyl groups in an aqueous solution, protons are more likely to be lost. Whereas, in a non-aqueous solution, intramolecular hydrogen bonds are more likely formed, hydrogen atoms are more likely to be lost.

### 3.2. Calculation Study of Antioxidant Activity of Anthraquinone Compounds with Phenolic Substituents

A density functional theory approach has been successfully used to calculate the biochemical parameters of polyphenol extracts and to elucidate the structure-activity relationship of phenolic antioxidants. Reza Nazifi et al. [58] predicted the antioxidant activity of three aloe compounds (aloe-emodin (**2**), aloesone (**2a**) and isoeleutheol (**2b**)) by using the density functional theory calculation of the B3LYP hybrid functional and the 6-311++G** basis set. HOMO-LUMO energy gaps showed that compound **2** had the lowest *E*^gap^ value (Appendix A), so it was more prone to give electrons and showed the best antioxidant activity among the compounds **2**, **2a**, and **2b** (Figure 3). From the structures of them, it clearly showed that compound **2** had more electron-donating phenoxy groups.

Jeremic et al. [59] used the M06-2X/6-311++G(d,p) theory to study the antioxidant capacity of compounds **1**, **2**, **5**, **6**, **10** and **11**. The results showed that compounds **10** and **11**, with the hydroxy groups on the *ortho* position, had the highest reactivity. Compounds **1** and **6** with three hydroxy groups had moderate antioxidant capacity. Compounds **2** and **5** with two hydroxy groups had the lowest antioxidant capacity. Isin [60] used B3LYP/6-3111++G(2d,2p) and the conductor-like polarizable continuum model to study the free radical scavenging activities of four hydroxyanthraquinones (compounds **11**, **12**, **13** and **14**). The order of hydrogen supply capacity is compounds **11** > **12** > **14** > **13**, and the BDE values of compounds **11** and **12** (*ortho*-hydroxy groups) are lower than the BDE values of compounds **13** and **14** (carboxyl group).

It was found that 1-OH and 8-OH are the most active sites of compounds **2** and **5**, while 3-OH is the most active site for compounds **1** and **6** [61]. Based on the theoretical level of B3LYP/6-311++G**, Markovic et al. [62] analyzed the BDE values of all hydroxyl positions of compound **1** and clarified the role of 3-OH in antioxidant properties. The BDE values of 1-OH and 8-OH are higher because H removal also means the fracture of hydrogen bonds (Appendix A). These results can be explained by the intramolecular hydrogen bonds formed between 1-OH/8-OH and 9-carbonyl group, which make the hydrogen atom difficult to be lost.

Jeremic et al. [63] evaluated the antioxidant activity of compound **10** by its BDE and IP theoretically (Appendix A). The free radical scavenging characteristics of **10** in the gas phase were well explained by HAT. The lower IP value for aqueous solution indicated that the SET-PT mechanism was reasonable.

## 4. Antioxidant Experiments of Anthraquinone Compounds In Vitro

In terms of antioxidant properties, anthraquinone compounds can scavenge ^•^OH, 1,1-diphenyl-2-picrylhydrazyl (DPPH^•^) and O_2_^•−^ free radicals and inhibit lipid peroxide to exert their antioxidant effects, which is of great significance for the development of high efficiency antioxidants. In this section, the antioxidant properties of natural and synthetic anthraquinone compounds are discussed in detail according to scavenging activities of different radicals.

### 4.1. Antioxidant Experiments of Natural Anthraquinone Compounds

#### 4.1.1. Study on Scavenging Activities of ^•^OH Radical

Jung et al. [64] found that compound **9** had a stronger inhibitory effect on ^•^OH than compound **1**, with IC_50_ values of 3.05 ± 0.26 µM and 13.29 ± 3.20 μM, respectively. In addition, compounds **1** and **9** exhibited hepatoprotective effects against tacrine-induced cytotoxicity. Vargas et al. [65] investigated the ability of compounds **1**, **2** and **3** to inhibit ROS (^•^OH, ^1^O_2_, H_2_O_2_) in cell-free systems. The results showed that the scavenging ability was in the following order: compounds **1** > **3** > **2**. Kumar et al. [66] found that compound **11** was an extraordinarily fine scavenger of ^•^OH, which in turn protected plasmid DNA from damage. Lin et al. [67] found that the reason why Folium *Sennae* could protect against DNA and mesenchymal stem cell damage induced by ^•^OH. It might be because plant phenols (especially compounds **1**, **2**, **3**) played a protective role through HAT and/or SET-PT mechanisms. Their phenolic hydroxyl groups were partially oxidized into stable semiquinone forms. The stability of the semiquinone form ultimately determined the protective or antioxidant effect of plant phenols.

#### 4.1.2. Study on Scavenging Activity of DPPH^•^ Radical

It was reported that compound **1** and **15** could significantly eliminate DPPH^•^ radicals and that the scavenging ability was dose-dependent at concentrations of 0.5~100 µM [23,68]. Both Nam et al. [69] and Baghiani et al. [70] found that compound **10** had a high radical scavenging effect with an IC_50_ = 3.491 μg/mL. The experimental data demonstrated that the number and site of the OH groups seemed to be the primary factors affecting the abilities of antioxidants. The *ortho*-hydroxyl groups in compound **10** can react with free radicals to form a more stable conjugated semiquinone free radical and thus interrupt the free radical chain reaction. Therefore, compound **10** had stronger antioxidant properties. Shi and Huang [71] found that compound **5** had a scavenging effect on DPPH^•^ radicals (IC_50_ = 26.56 µg/mL). Zengin et al. [22] estimated the free radical scavenging effects of compounds **10** and **11** through ABTS and DPPH assays. The results showed that compound **10** had the greatest scavenging activity, followed by compound **11**.

In general, scavenging of ^•^OH radicals is often proceeded via a radical adduct formation mechanism. First, a radical adduct with phenol is formed. Then, water molecules are eliminated [51]. Scavenging of DPPH^•^ is often based on the abstraction of hydrogen atoms from antioxidants or the transfer of electrons from phenoxide anions to DPPH^•^ [72].

#### 4.1.3. Study on Scavenging Activity of O_2_^•−^ Radical

Our group found that compound **1** showed weaker O_2_^•−^ scavenging capacity (IC_50_ = 235.73 μM) than Vitamin C (IC_50_ = 12.68 μM) using pyrogallol autoxidation. The mechanism for scavenging of O_2_^•−^ might be by electron donation from the reductants, so Vitamin C had better ability [73].

#### 4.1.4. Determination of Anti-Lipid Peroxidation

The lipid peroxidation reaction is shown in Appendix A. Lipid peroxidation is one of the results of free radical formation in cells and tissues. O_2_^•−^, H_2_O_2_ and ^•^OH actively participate in the initiation of lipid peroxidation.

Yen et al. [74] reported that anthraquinone compounds inhibited the peroxidation of linoleic acid, and the order of the activity was anthrone compound > **2** > **3** > **1** > anthraquinone. Although the inhibitory activities of anthraquinone compounds were weaker than anthrone, which may be attributed to the keto carbonyl group of C-10, other reports showed that the above anthraquinone compounds were good antioxidants against lipid peroxidation [75,76,77]. For example, the inhibitory effect of compound **1** was even stronger than that of α-tocopherol. The activities could be tested with three kinds of systems: (1) in linoleic acid; (2) 2′,7′-dichlorodihydrofluorescein diacetate (DCHF-DA); (3) 3-morpholinosydnonimine (SIN-1) system. The inhibitory effect of anthraquinone compounds might be related to radical scavenging, iron chelation, and enzyme affinity.

#### 4.1.5. Natural Anthraquinone Compounds in Cell Culture-Based Experiments

Compared with other in vitro experiments, cellular experiments are performed in an environment that more closely resembles the human physiological environment, and some of the results are listed in Table 1. The results from cellular experiments showed that anthraquinone compounds may have a series of beneficial bioactivities, such as oxidative stress reducing, cytotoxicity prevention, and inflammatory inhibition. Some of the results are listed as followings:

Compound **1** could fight cisplatin-induced oxidative stress in human embryonic kidney 293 (HEK 293) cells by restoring the GSH and total antioxidant capacity (TAC) depletion as well as augmenting the cisplatin-inhibited antioxidant enzymes, such as SOD, CAT, GPX, GR, and GST [68].Compound **2** could prevent H_2_O_2_-induced cytotoxicity of PC12 cells by significantly reducing both the extracellular release of NO and lactate dehydrogenase and intracellular accumulation of ROS [78]. Compound **3** could obviously increase the viability of H_2_O_2_-injured human umbilical vein endothelial cells (HUVECs) by downregulating malondialdehyde (MDA), lactate dehydrogenase (LDH) and the expression of caspase-3, caspase-8, caspase-9 mRNA, while upregulating the NO content, nitric oxide synthase (NOS), SOD, and glutathione peroxidase (GSH-PX) [79]. Compound **5** could inhibit lipopolysaccharide (LPS)-induced inflammatory reaction of BV-2 mucin microglia by downregulating dynamin-related protein 1 (Drp1) (S637), mitogen-activated protein kinase (MAPK), nuclear factor-kappa B (NF-κB) and ROS generation [80]. Compound **10** could ameliorate alcohol-induced hepatotoxicity by reducing excess ROS generation, promoting nuclear factor erythroid-2 related factor 2 (Nrf2) expression and enhancing hepatic antioxidant defense systems [81]. Kim et al. also determined that compound **10** could reduce neuronal damage and inflammatory responses after oxidative stress in HT22 cells or ischemic damage in gerbils via MAPKs, Bax, and oxidative stress cascades [82].

### 4.2. Antioxidant Experiments of Synthetic Anthraquinone Compounds

#### 4.2.1. Anthraquinone Metal Complexes and Antioxidation Activities

Metal complexes often have excellent antioxidant activity and are useful for diseases related to oxidative stress. Many studies have reported that the combination of the active ingredients of traditional Chinese medicine and metal ions may play a synergistic role. Compared with ligands, metal complexes often have enhanced pharmacological activities or produce some new pharmacological effects and may also reduce toxicity and side effects. Our group found that metal complexes had higher antioxidant activity, which may be due to the synergistic effect of metals and ligands [73,83,84,85,86,87]. Several reports studied the antioxidative activities of the metal complexes of **1**, such as Cu(II), Fe(II), Zn(II), Mg(II) and Mn(II). The results showed that all of them have higher antioxidant activities than the ligand. Compound **1** formed the metal complexes through the coordination of the 9-carbonyl group and 1-phenolic hydroxyl group and in the structures of the complexes, the O atoms of 1-hydroxyl and 9-carbonyl coordinated with metal ions to form a stable hexacoordinated ring structure [88,89,90].

In the case of metal complexes, the active center of the potential reaction is transferred from the hydroxyl group to the metal ion. The C=O and phenolic hydroxyl groups in the anthraquinone structure are the active sites with the electron donor before coordination. For metals, the merged orbitals can be split into different energy levels when coordinating with ligands. After absorbing the electromagnetic wave energy, the electrons can jump from the low-level d orbital to the high-level d* orbital and generate absorption bands in the visible region. As shown in Figure 4, the 1-hydroxyl group and 9-carbony1 group of anthraquinone compounds can provide metal chelation sites. They have a completely large π-conjugated system, strong coordination of oxygen atoms and an appropriate spatial configuration, so they can be used as good chelating ligands for metal ions to coordinate with metal ions and form complexes [50,73].

#### 4.2.2. Chemical Modification of Anthraquinone Compounds

Taking anthraquinone as the lead compound, it is of great practical significance to study its structural modification to enhance its antioxidant activity. The structural characteristics of the substituents have an important effect on the scavenging activity of a single molecule. This effect is reflected mainly in the possibility of stabilizing the free radicals [91].

As shown in Figure 5, compounds **17a** and **18** have different parent structures with the same substituent group. It could be found that the antioxidant capacity of compound **18** was higher than that of compound **17a**. The possible reason was that the electronegativity of S atom was lower than that of O atom. The reducing capacity was stronger, the antioxidant activity was higher. As for the compounds **17a**–**17e**, they had the same parent structure and different R groups. The antioxidant assay showed that the chain length was related to the antioxidant activity when comparing compounds **17a** and **17b**. When phenyl substitution (compounds **17d** and **17e**) was introduced, the antioxidant activities were increased due to the conjugation system formed between phenyl group and the parent structure. However, compound **17c** had a relatively lower inhibition which may be due to the breaking of conjugation structure caused by the steric hindrance effect of the *ortho* substitution [92].

## 5. In Vivo Antioxidant Experiments with Anthraquinone Compounds

Anthraquinones have a wide range of biological activities and can be used to treat many kinds of diseases, such as Alzheimer’s disease, inflammation, cancer, liver injury, diabetes, gastrointestinal disorders (e.g., diarrhea, constipation and dysentery), ulcers, radiation injury, and burns [93,94,95]. Especially, anthraquinones have remarkable abilities in scavenge free radicals and prevent oxidative damage to tissues [96]. Although the specific relationship between the antioxidant mechanism and the antioxidant activity in vivo has not been clarified, some information can be obtained from the bioactivity and pharmacokinetic studies. In general, the mechanism for enhancing the activity of antioxidant enzymes may relate to the redox activity of anthraquinones or their ability to bind specific proteins or both. Some of the in vivo antioxidant activities of anthraquinones are listed in Table 2.

### 5.1. Pharmacological Activities of Anthraquinone Compounds

#### 5.1.1. Anti-Neurodegenerative Diseases

Oxidative stress is an early event in the development and progression of Alzheimer’s disease [97]. As seen in Alzheimer’s disease patients, oxidative stress contributes significantly to the perturbation of calcium homeostasis and subsequent apoptosis [98]. Many clinical studies have reported strong evidence of the involvement of oxidative stress in the pathogenesis of Alzheimer’s disease [99]. The Aβ aggregation inhibition assay of compound **1** showed that compound **1** could block Aβ_42_ fibrillogenesis and Aβ-induced cytotoxicity [100]. Chen et al. [101] found that compound **1** could protect nerve cells, normalize synaptic damage by reducing the phosphorylation of the extracellular signal-regulated kinase 1/2 (ERK1/2), decrease ROS and protect mitochondrial function. Tao et al. [78] found that compound **2** could show an important role in cognitive deficits in a scopolamine-induced amnesia animal model. In addition, compound **2** increased SOD, GPX and the content of acetylcholine but decreased the level of MDA and acetylcholinesterase activity. Therefore, the results indicated that compound **2** might have a neuroprotective effect on Alzheimer’s disease by inhibiting the activity of acetylcholinesterase and regulating oxidative stress. In a mouse middle cerebral artery occlusion (MCAO) model, Zhao et al. [102] found that compound **5** reduced neuronal damage related to nitric oxide production by reducing the expression of cleaved caspase-3 and enhancing the activity of SOD and manganese-dependent SOD. Zhang et al. [103] found that compound **5** could alleviate hippocampal neuronal injury in lead-exposed neonatal mice, while significantly alleviate the level of MDA in the brain, kidney, and liver and increase the activity of SOD and GPX.

#### 5.1.2. Anticancer

Previous reports showed that the oxidative stress level in cancer cells was higher than that in healthy cells. This stress is accompanied mainly by the production of higher levels of free radicals, causing DNA degradation leading to the carcinogenesis of normal cells and their transformation into cancer cells [104]. In addition, reducing nuclear factor erythroid 2-related factor 2 signaling and strengthening the inflammatory pathway through ROS production are related to carcinogenesis [105,106]. Compound **1**/cisplatin combination therapy inhibited the growth of human ovarian carcinoma cells and gallbladder carcinoma cells in vivo. The mechanism may involve the downregulation of multidrug resistance-related protein (MRP1) expression [107] and ATP-binding cassette superfamily G member (ABCG2) expression [108]. Wang et al. [109] found that compound **1** could induce necroptosis through ROS-mediated activation of the c-Jun N-terminal kinases (JNK) signaling pathway and also inhibit glycolysis by downregulating GLUT1 through ROS-mediated inactivation of the phosphoinositide 3-kinases (P13K)/AKT signaling pathway.

#### 5.1.3. Anti-Hepatopathic Diseases

Oxidative stress is also a pathogen in liver inflammation that can lead to fibrosis and hepatocellular carcinoma [5]. Emodin and Rubiadin (compounds **1** and **16**) are well-known anthraquinone compounds whose hepatoprotective activity has been widely studied [24,110,111,112]. The results showed that compounds **1** and **16** could inhibit ROS generation and mitochondrial damage through the AMP-activated protein kinase (AMPK)/Yes-associated protein (YAP)-mediated pathway in a hepatocyte cell line and decrease acetaminophen (APAP)-induced oxidative damage in mice. Increases in the levels of GSH, glutathione reductase (GRD), GPX and GST by treatment with compound **1** were associated with a decrease in the level of MDA, indicating a decrease in oxidative stress.

#### 5.1.4. Anti-Cardiovascular Diseases

Excessive ROS may damage vascular cells, induce the proliferation and migration of vascular smooth muscle cells, and then cause vascular remodeling [6,7]. Emodin (compound **1**) showed a potential protective effect on cardiovascular system and the mechanism was connected with its antioxidation activities. The antioxidation activities might be related to the inhibition of the contractile effect of 5-hydroxytryptamine and synergizing the diastolic effect of acetylcholine related to up-regulation of free radicals, hydrogen peroxidation and cGMP [7,113].

#### 5.1.5. Anti-Inflammatory Activity

The inflammatory response produces a large amount of ROS and is related to many kinds of diseases. The NF-κB transcription factor family has been considered the central mediator of the inflammatory process and participates in adaptive and innate immune responses [114]. A large number of studies have shown that ROS or metals can affect nuclear NF-κB transcription factors [115]. Lipid peroxidation has been reported to occur in acute pancreatitis [116,117]. ROS can activate NF-κB, which can in turn adjust the expression of inflammatory cytokines [118].

Rat models of severe acute pancreatitis (SAP) was used to analyze the molecular mechanisms and effect of compound **1** [119,120]. As a natural antioxidant, compound **1** could inhibit NF-κB activation and regulate the production of cytokines, such as tumor necrosis factor (TNF)-α, interleukin (IL)-1β, and IL-6, and sequentially regulate the oxidative stress response. Song et al. [121] used proteomics to investigate the antioxidant mechanism of compound **1** on the liver of large herring. Compound **1** was found to significantly increase the expression of antioxidant-related mRNAs, such as GPX-1, GST and heat shock 70 kDa protein (HSP70), compared with the control group. mRNA expression of sorbitol dehydrogenase (SORD) and glyceraldehyde-3-phosphate dehydrogenase (GAPDH) decreased, which was consistent with protein expression. There were also reports about the protection activity of compound **1** from cigarette smoke-induced lung inflammation and oxidative damage. Xue et al. [122] found that compound **1** significantly attenuated the expression of TNF-α, IL-6 and IL-1β and oxidative stress by enhancing the activities of SOD, CAT and GPX in mice with acute pulmonary inflammation induced by cigarette smoke. The results of anti-neuroinflammatory activities showed that compound **2** protected against brain impairment, mainly attributed to the antioxidant activities of compound **2** via the P13K/AKT/mTOR and NF-κB activation [123]. Compound **5** was also found to have good hepatoprotective activity, which was expressed by significantly reducing of inducible nitric oxide synthase (iNOS), TNF-α, IL-6 and IL-10 through inhibiting the expression of NF-κB and receptor-interacting protein 140 (RIP140) in LPS-induced acute liver injury in mice [124].

**Table 2 molecules-28-08139-t002:** In vivo antioxidant activities of anthraquinone compounds.

Comp.	Model ^a^	Major Discoveries and Proposed Mechanisms ^b^	Dosage of Administration	Refs.
Emodin (**1**)	C57B/6 mice orally injected with APAP	↓ Oxidative damage	10, 30 mg/kg	[110]
	CCl_4_-intoxicated mice	↑ GSH, GRD, GPX, GST ↓ MDA ↓ Oxidative stress	2.6 mg/g	[112]
	SAP-induced mice	↑ SOD ↓ NF-κB, TNF-α, IL-6, IL-1β ↓ MDA ↓ Oxidative stress	1 mg/kg	[120]
	SAP-induced mice	↑ VDAC1 ↓ ROS ↓ Serum amylase, lipase, TNF-α, IL-18, caspase-1, NLRP3	6 mg/mL	[119]
	Female C57B1/6 mice	↑ SOD, CAT, GPX ↓ TNF-α, IL-6, IL-1β	40 mg/kg	[122]
	Healthy *M. amblycephala* fingerlings	↑ GPX1, GSTm, HSP70 ↓ GAPDH, Sord	30 mg/kg	[78]
	6-week-old BALB/c-nu/nu mice	↓ ABCG2 expression	20, 40, 60 mM	[108]
Aloe-emodin (**2**)	Scopolamine-induced amnesia animal model	↑ SOD, GPX, ACh ↓ MDA, AChE	IC_50_ = 18.37 μg/mL	[78]
	Male C57BL mice	↑ SOD, MnSOD ↓ Cleaved caspase-3	0.1–10 mg/kg	[102]
Chrysophanol (**5**)	Male BALB/c mice	↑ SOD, GPX, GSH, CAT ↓ MDA ↓ Oxidative stress ↓ TNF-α, IL-6, IL-10, iNOS, NF-κB, RIP140	1, 10 mg/kg	[124]
	Lead poisoned Kunming mice	↑ SOD, GPX ↓ MDA	10.0 mg/kg	[103]

^a^ Cell lines: APAP (acetaminophen), CCl_4_ (carbon tetrachloride), SAP (severe acute pancreatitis) ^b^ “↑” upregulating or activation. “↓” downregulating or inhibition.

### 5.2. Pharmacokinetic Studies of Anthraquinone Compounds In Vivo

Pharmacokinetic studies, as an important part of preclinical and clinical studies during the development of innovative drugs, provide a theoretical basis for guiding the clinical and rational use of drugs, predicting drug interactions, understanding drug action mechanisms, exploring new modification methods for natural products, and designing drug delivery systems, etc. [125]. Since traditional Chinese medicine is mostly administered orally, the metabolism of anthraquinones contained in traditional Chinese medicine will first enter the absorption phase. The absorption of anthraquinone compounds depends on their physicochemical properties. The main site of absorption of anthraquinones is the intestine, not the stomach. The absorption of anthraquinone is related to the weak acidity of anthraquinones and the pH conditions in the intestine. When placed in a weakly acidic environment, since most anthraquinones are weakly acidic, resulting in lower ionization. In contrast, at higher pH values, the degree of ionization of anthraquinones increases, resulting in little absorption of anthraquinones [126].

In general, the pharmacokinetic data of anthraquinones are in a wide range in vivo. This may be due to differences in drug dosages, detection instruments, and protocols. As described in Appendix A, compound **3** had the minimum peak time (T_max_), the highest peak concentration (C_max_) and area under the curve (AUC) in dogs. A large number of pharmacokinetic studies have shown that compound **3** is the main component absorbed into the blood after gastrointestinal administration of rhubarb and its compound preparation containing rhubarb in both rats and humans. The peak concentration of oral absorption and bioavailability of compound **3** are superior to other free anthraquinones contained in rhubarb [127,128,129,130]. The water solubility of compound **3** may be improved due to its carboxyl group. A better lipid water distribution coefficient is favorable for compound **3** penetrate the amphiphilic biofilm, leading to the best bioavailability.

Biotransformation is a significant process during which anthraquinones are converted into inactive or more active metabolites and removed from the body. The transformation occurs mainly in the liver. Phase II conjugates, in particular glucuronides of rhubarb anthraquinones, were considered to be the predominant form in vivo [131,132]. The metabolic pathways and metabolites of anthraquinones are listed in Table 3.

Xu et al. [133] found that in a rat liver microsome incubation system, both compounds **1** and **5** were hydroxylated. Compounds **2** and **5** could be converted to compound **3** by oxidation. Compound **5** was transformed to demethylated forms and acetylated forms as dihydroxy-**5**, while compound **4** was transformed to demethylation forms as compound **1** isomer. Notably, the demethylation of compound **4** may be the reason for the low bioavailability of **4** [134]. Song et al. [132] separated and identified the metabolites in rat urine, bile and plasma after oral administration of rhubarb decoction and found that compounds **1**–**5** might be metabolized to sulfonated forms. Compound **2** was transformed to **2**-1-O-glucoside-8-O-glucuronide or **2**-8-O-glucoside-1-O-glucuronide, 2-hydroxyaloe-**1**-ω-O-glucuronide through hydroxylation, hydrogenation, glucuronidation and oxidized to compound **3** by oxidation. The oxidation reaction increases the bioavailability of anthraquinones. The order of bioavailability of anthraquinones was compounds **3** > **1** > **5** > **2,** which may result from compounds **2** and **5** being oxidized to compound **3** [135]. In addition, compounds **1**–**5** could be metabolized to sulfonated forms [132], which may result in a decrease in the oral bioavailability of anthraquinones [131].

Note that anthraquinones are not only phenols but also *p*-quinones, and the oxidation process of phenols also involves different forms of quinones. As such, quinones undergo enzyme-catalyzed one- or two-electron reduction reactions generating the corresponding semiquinone or hydroquinone, respectively. As shown in Appendix A, semiquinone, generated through the one-electron reduction pathway, undergoes rapid autoxidation under atmospheric conditions, transferring excess electrons to molecular oxygen, thereby generating O_2_^•−^ and regenerating the parent quinone. The redox cycling of enzyme-catalyzed reduction of quinone and the aerobic oxidation cycle of the semiquinone producing O_2_^•−^ continue until the system becomes anaerobic. At this time, the production of O_2_^•−^ is reduced, and the accumulation of semiquinone occurs. The result of this redox cycle is oxidative stress. Under anaerobic conditions, semiquinone is disproportionate to quinone and hydroquinone. However, hydroquinone from the two-electron reduction pathway, depending on its stability, may be discharged through four different pathways: (1) the detoxification pathway; (2) oxidized by O_2_ one electron at a time to produce O_2_^•−^ and the semiquinone; (3) forming reactive alkylating agents by a rearrangement reaction; and (4) undergoing a comproportionation reaction with quinone to produce semiquinone [136].

After oral administration, natural anthraquinones are absorbed into the blood by the intestinal mucosa, combined with glucuronic acid or sulfuric acid in the liver and intestinal tract, and then transported by blood circulation to various tissues and organs throughout the body to exert a variety of pharmacological effects. Anthraquinones undergo oxidation, methylation, esterification and glycosidation reactions when being metabolized. Anthraquinones have good biological activity in vitro, but due to their low solubility and rapid elimination rate, the oral absorption utilization ratio is low. In addition, side effects of anthraquinones have also been reported, including genotoxic [137], nephrotoxic [138], and hepatotoxic effects [139]. Therefore, it is necessary to modify the structure, add absorption promoters, or change drug dosage forms to improve the bioavailability of anthraquinones. Further systematic studies on the absorption and metabolism of anthraquinone compounds in vivo on the basis of pharmacology are needed to ensure their safety and effectiveness in clinical applications.

## 6. Analysis of the Structure-Activity Relationship

The following three structural characteristics of anthraquinone compounds are closely related to their antioxidant activity: (1) the benzene ring and carbonyl group in the parent structure; (2) the number and position of the hydroxyl groups; (3) the position and number of other substituents, such as carboxyl group, methyl group and methoxyl group. The benzene ring in anthraquinone compounds is hydrophobic, while the phenolic hydroxyl group is hydrophilic. As shown in Figure 1, anthraquinone compounds can be roughly divided into two types according to their structure: emodin type and alizarin type. The hydroxyl groups of the emodin type, such as compounds **1**–**9**, are distributed on both sides of the benzene ring. Hydroxyl groups of the alizarin type are distributed on the benzene ring on one side, such as compounds **10**–**12**.

The antioxidant activity of anthraquinones is mainly related to phenolic hydroxyl groups. There are three main types of phenolic hydroxy-related antioxidant mechanisms. The first is the HAT mechanism, in which the hydrogen atom is extracted from ArOH. So, BDE of the O-H bond is a significant parameter for scavenging activity evaluation. The second mechanism is SET-PT, namely, the rate determining step is losing one electron from ArOH to generate ArOH^+•^ and then followed by a proton transfer step. Therefore, IP is the most important parameter for evaluating antioxidant activity. The third mechanism is SPLET, which means a proton is first lost to form ArO^−^, and then electron transfer occurs. Therefore, p*K*_a_ of ArO-H bond is main determining factor in this mechanism.

The experimental results also supported these mechanisms and the results will be discussed according to the following structural characteristics: (1) The position and the number of the phenoxyl groups (Appendix A) [54,59,60]: the distribution of hydroxyl groups on the ipsilateral benzene ring improved the free radical scavenging activity, which may be related to the *o*-diphenol structure (e.g., compounds **9**, **10**, **11**). In this kind of structure, the formation of intramolecular hydrogen bonds leads to lower BDE values of OH bonds and lower p*K*_a_ of ArO-H, and thus higher antioxidative activity, which accord to both HAT and SPLET mechanism; (2) The position of other substitutions: When the carboxyl group is on the *ortho* position to the hydroxyl group (compounds **12** and **13**) or the methoxyl group is on the meta position (compound **4**), they are the electron withdrawing groups. The electron density on the hydroxyl group decreases and the electron is difficult to lose with a higher IP value. The antioxidant activity decreases, which is consistent with the mechanism of SET-PT; (3) The carbonyl group on the parent structure will also affect the free radical scavenging activity. It worth to mention that 1-OH/8-OH group and carbonyl group can work as the chelating sites to coordinate with Fe(II) or Cu(I) ions, which may terminate the Fenton reaction. Furthermore, natural compounds with similar structure have antioxidative activities following this path, which has been proved by the research results from our group [73,83,84,85,86,87,88].

## 7. The Neutraceutical Properties of Natural Anthraquinone Compounds for Clinical Application

Anthraquinones are widely distributed in various botanicals, such as rhubarb, aloe and Fo-Ti [140,141], which are commonly clinically used in traditional Chinese medicines and dietary supplements. As shown in Table 4, various levels of compounds **1**, **2** and **3** have been reported in three botanicals, rhubarb, Fo-Ti and aloe. The daily intake levels of the three botanicals in traditional Chinese medicines and dietary supplements were also identified. In addition, the diseases that could be treated by these botanicals were listed in Table 4.

## 8. Conclusions and Future Perspectives

Natural anthraquinones are a class of natural products with important biological activities, commonly found in many plants and microorganisms. These compounds have been extensively studied and proven to be promising for a wide range of applications in many fields, one of which is as antioxidants. In recent years, numerous studies have shown that natural anthraquinones possess strong antioxidant activity. As shown in this review, anthraquinones (1) have a significant free radical scavenging ability, (2) can modulate complex antioxidant enzymes and non-enzymatic systems to reduce oxidative stress, and (3) inhibit the formation of ROS or inhibit enzymes involved in free radical generation by chelating trace elements. Therefore, natural anthraquinones have a broad application prospect in the field of antioxidants. However, more in-depth research is needed in the following two aspects to make anthraquinones more reliably and widely used in medicine in the future.

First, although drugs based on anthraquinone compounds, such as doxorubicin, mitoxantrone, and idarubicin, etc., have been used successfully to treat diseases and the anthraquinone core may continue to be a potential scaffold for developing novel therapeutic candidates, side effects and low absorption of anthraquinones have also been reported. Therefore, further systematic studies in relation to its SAR as well as the absorption and metabolism in vivo are needed to accelerate the development and utilization of anthraquinone compounds as promising drug candidates in the future. And in this regard, a brief summary of SAR studies on anthraquinone compounds was provided in Section 6 of this review. At the same time, it is also necessary to optimize the structure of anthraquinone compounds by chemical modification. In this review, the summary of the properties of some synthetic anthraquinone compounds was reported in the Section 4.1 and recently our group also reported a series of anthraquinone compounds with tertiary amine substituents with promising anti-AD activities [73].

Second, although antioxidants are essential for human health, due to the specific complicated activities of ROS, drugs that regulate the imbalance between the formation and elimination of ROS should be developed more rationally to provide new treatments for various diseases mediated by ROS. After in-depth literature research, we believe that more detailed studies on the mechanism are needed in the future, especially in the following two kinds of relationship. One is about the relationship between the concentration of the anthraquinone compounds clinically applied and their ROS mediation properties. As shown in the review, most of literature reported anthraquinone compounds’ inhibition of the expression of ROS via SOD, H_2_O_2_, superoxide and NF-κB pathway. However, some groups reported that anthraquinone of high concentration may alter the subcellular redox equilibrium and produce cytotoxicity induced by ROS. For example, emodin (1–25 μg·mL^−1^) induced mitochondria-induced cell apoptosis associated with generation of ROS [7,121]. So, in Section 7 of this review, botanical daily intake (g/day) values were given for the neutraceutical application of natural anthraquinone compounds. The other one is about the relationship between scavenging of ROS and the treatment of diseases. Most time, decreasing of ROS is needed for the treatment of diseases with anthraquinone compounds. However, there were reports about elevation of ROS in the process of disease treatment. For example, Anthraquinones [149], were reported to increase ROS and reduce the activities of antioxidant enzymes in cells, and thus induce autophagy and apoptosis in cancer cells.

The results of this work can be applied to further understand the antioxidant mechanism of anthraquinone compounds and it also has guiding significance for the study of antioxidant ability of other natural products with similar structure.

## Figures and Tables

**Figure 1 molecules-28-08139-f001:**
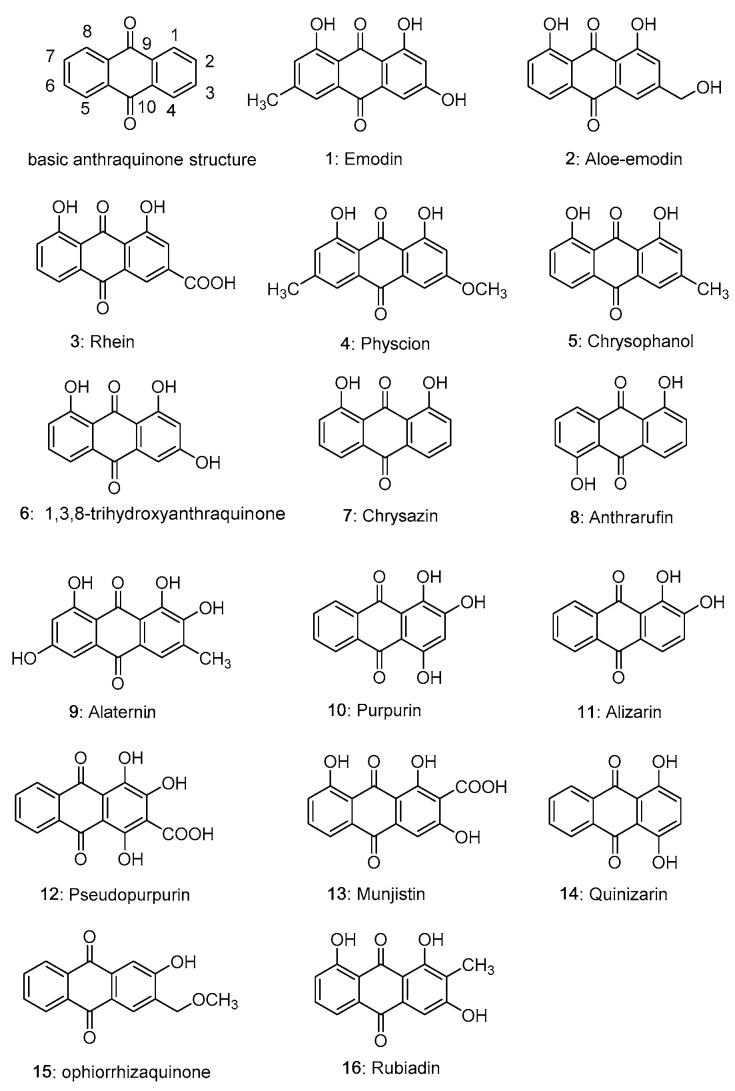
Structures of common natural anthraquinone compounds.

**Figure 2 molecules-28-08139-f002:**
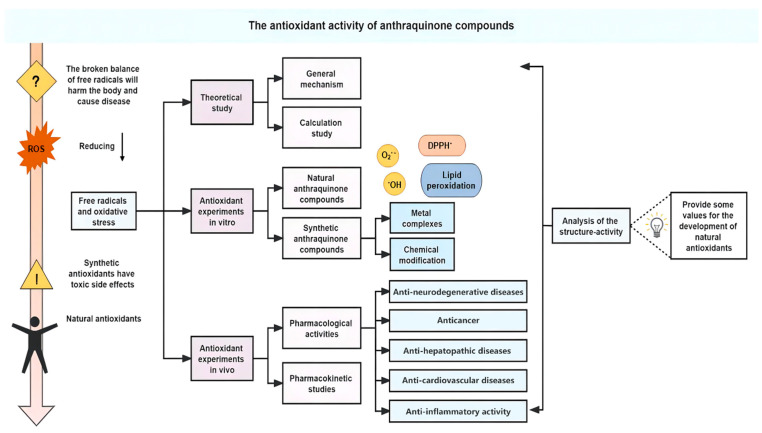
The main line of this work.

**Figure 3 molecules-28-08139-f003:**
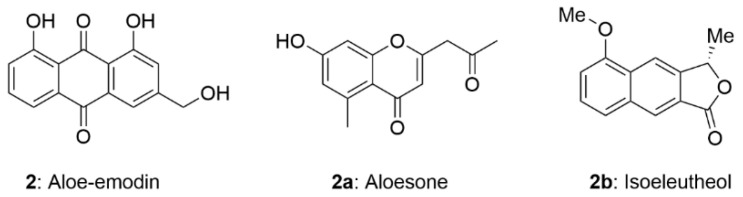
The structure of aloesone, aloe-emodin and isoeleutheol.

**Figure 4 molecules-28-08139-f004:**
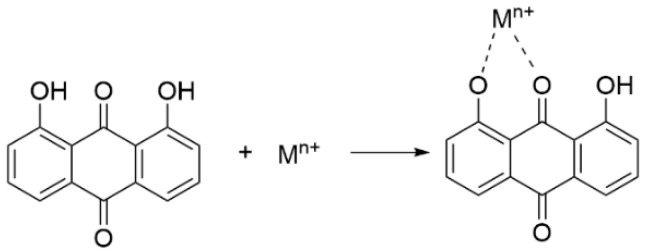
The chelation of anthraquinone and metals.

**Figure 5 molecules-28-08139-f005:**
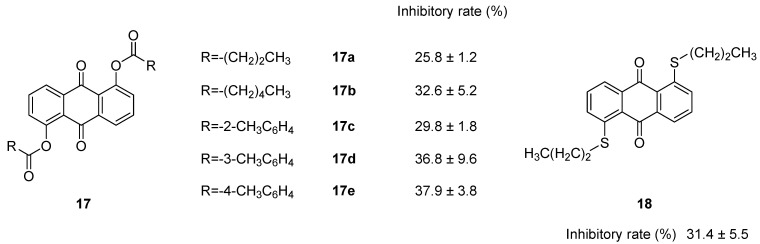
Summary of the effects of 1,5-bisacyloxy-anthraquinones with O–linked substituents (**17**, **17a**–**11e**) and 1,5-bisacyloxy-anthraquinones with S–linked substituents (**18**) on superoxide production in human neutrophils. Data are expressed as inhibition (%) at 100 μM. Values represent the means ± s.e.m. of 3–6 experiments performed on different days using cells from different donors.

**Table 1 molecules-28-08139-t001:** Antioxidant activities of anthraquinone compounds with cellular experiments.

Comp.	Model	Major Discoveries and Proposed Mechanisms ^a^	Dosage of Administration	Refs.
Emodin (**1**)	human embryonic kidney cells (HEK 293)	↑ SOD, CAT, GPX, GR, GST	0.5 μM	[68]
Aloe-emodin (**2**)	H_2_O_2_-induced PC12 cells	↓ AChE ↓ Oxidative stress	-	[78]
Rhein (**3**)	H_2_O_2_-induced human umbilical vein endothelial cells	↑ NO, NOS, SOD, GSH-PX ↓ Caspase-3, -8, -9 mRNA, MDA, LDH	2, 4, 8, 16 μM	[79]
Chrysophanol (**5**)	BV-2 mucin microglia	↓ Drp1 (S637) ↓ MAPK, NF-κB, ROS	10 μM	[80]
Purpurin (**10**)	primary hepatocytes and WRL-68 cells	↑ Nrf2, GST, GPX, GR, CAT, SOD	30, 100 μM	[81]

^a^ “↑” upregulating or activation. “↓” downregulating or inhibition.

**Table 3 molecules-28-08139-t003:** Metabolic pathways and metabolites of anthraquinones.

Comp.	Models	Dosage of Administration	Metabolic Pathway	Metabolites	Refs.
Emodin (**1**)	Sprague-Dawley rats	10 mL/kg rhubarb decoction	glucuronidation oxidation hydroxylation hydrogenation sulfation	emodin-*O*-diglucuronides, emodin-*O*-glucoside-*O*-glucuronide, 1,8-dihydroxy-3-carboxy-6-methylanthraquinone-1 or 8-*O*-glucoside, emodin-1 or 8-*O*-glucuronide-3-*O*-sulfate or emodin-1 or 8-*O*-sulfate-3-*O*-glucuronide, 1,3,8-trihydroxy-6-methyl-10-oxanthranol glucuronide, emodin-*O*-diglucuronides, 1,3,8-trihydroxy-6-(glucuronidyl)methylanthrquinone, emodin acid-*O*-glucuronide, emodin-2-C-glucuronide, emodin-3-*O*-glucuronide	[132]
	Sprague-Dawley rats	0.0156 mg/mL	transhydroxylation hydroxylation reduction oxidation dihydroxylation	hydroxy-emodin, dihydroxy-emodin, hydroxy-aloe-emodin, hydroxy-rhein, aloe-emodin isomer, aloe-emodin, emodin, 1,3,8-trihydroxy-6-methyl-9-oxanthranol/1,3,8-trihydroxy-6-methyl-10-oxanthranol	[133]
Aloe-emodin (**2**)	Sprague-Dawley rats	10 mg/kg rhubarb decoction	glucuronidation oxidation hydrogenation hydroxylation	aloe-emodin-8-*O*-glucoside-1-*O*-glucuronide or aloe-emodin-1-*O*-glucoside-8-*O*-glucuronide, 2-hydroxyaloe-emodin-ω-*O*-glucuronide	[132]
	Sprague-Dawley rats	0.035 mg/mL	hydroxylation reduction oxidation	dihydroxy-aloe-emodin, hydroxy-aloe-emodin, hydroxy-rhein, hydroxyl-1,8-dihydroxy-3-hydroxymethyl-9-oxanthranol/hydroxyl-1, 8-dihydroxy-3-hydroxymethyl-10-oxanthranol, aloe-emodin, rhein isomer	[133]
Rhein (**3**)	Sprague-Dawley rats	0.195 mg/mL	hydroxylation reduction	rhein, rhein isomer, dihydroxyl-1,8-dihydroxy-3-carboxyl-9-oxanthranol/dihydroxyl-1,8-dihydroxy-3-carboxyl-10-oxanthranol	[133]
Physcion (**4**)	Sprague-Dawley rats	0.16 mg/mL	demethylation hydroxylation reduction	emodin isomer, hydroxy-emodin, emodin, dihydroxy-1,8-dihydroxy-3-methoxy-6-methyl-9-oxanthranol/1,8-dihydroxy-3-methoxy-6-methyl-10-oxanthranol	[133]
	Sprague-Dawley rats	10 mg/kg rhubarb decoction	glucuronidation sulfation	physcion-1-*O*-glucoside-8-*O*-glucuronide or physcion-8-*O*-glucoside-1-*O*-glucuronide, physcion-1,8-*O*-diglucuronides	[132]
Chrysophanol (**5**)	Sprague-Dawley rats	10 mg/kg rhubarb decoction	glucuronidation sulfation	chrysophanol-1-*O*-glucoside-8-*O*-glucuronide, chrysophanol-8-*O*-glucoside-1-*O*-glucuronide, chrysophanol-1,8-biglucuronides, chrysophanol-1-*O*-glucuronide, chrysophanol-8-*O*-glucuroniede	[132]
	Sprague-Dawley rats	0.0755 mg/mL	hydroxylation acetylation demethylation, reduction, oxidation	dihydroxy-chrysophanol, chrysophanol, dihydroxyl-1,8-dihydroxy-3-methyl-9-oxanthranol/dihydroxyl-1, 8-dihydroxy-3-methyl-10-oxanthranol, hydroxy-chrysophanol, rhein	[133]

**Table 4 molecules-28-08139-t004:** Anthraquinone levels in botanical and botanical usage.

Comp.	Source Botantical	Botanical Daily Intake ^a^ (g/day)	Neutraceutical Properties	Refs.
**1** Emodin	Rhubarb (Da Huang, *Rheum officinale*, *Rheum palmatum*, *Rheum tanguticum*), root or rhizome	3–30 [141]	Tumor, Inflammation, Gastrointestinal disease, Hepatoprotective activity, Diabetic nephropathy, Atherosclerosis	[142,143,144]
	Fo-Ti (*Polygonum multiflorum*), root	3–12 [145]	Alzheimer’s disease, Parkinson’s disease, Hyperlipidaemia, Inflammation, Cancer	[146]
**2** Aloe-emodin	Rhubarb (Da Huang, *Rheum officinale*, *Rheum palmatum*, *Rheum tanguticum*), root or rhizome	3–30 [141]	Tumor, Inflammation, Gastrointestinal disease, Hepatoprotective activity, Diabetic nephropathy, Atherosclerosis	[142,143,144]
	Fo-Ti (*Polygonum multiflorum*), root	3–12 [145]	Alzheimer’s disease, Parkinson’s disease, Hyperlipidaemia, Inflammation, Cancer	[146]
	Aloe (*Aloe vera*, *Aloe barbadensis*), leaf	0.005–174 National Institutes of Health, (2019) [147]	Inflammation, Cancer	[148]
**3** Rhein	Rhubarb (Da Huang, *Rheum officinale*, *Rheum palmatum*, *Rheum tanguticum*), root or rhizome	3–30 [141]	Tumor, Inflammation, Gastrointestinal disease, Hepatoprotective activity, Diabetic nephropathy, Atherosclerosis	[142,143,144]
	Fo-Ti (*Polygonum multiflorum*), root	3–12 [145]	Alzheimer’s disease, Parkinson’s disease, Hyperlipidaemia, Inflammation, Cancer	[146]

^a^ Values used to calculate to estimate the daily intake of compounds **1**, **2** and **3**.

## Data Availability

Not applicable.

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
