# Peer review of "A Review on Bioactive Anthraquinone and Derivatives as the Regulators for ROS"

_molecules, 2023, doi:10.3390/molecules28248139_

Round 1

Reviewer 1 Report

Comments and Suggestions for Authors

The manuscript is definitely interesting. But I suggest you some points to improve the quality of this manuscript.

1. The manuscript lacks recommendation for future perspectives.

2. There is no description about clinical data of any compounds for antioxidant activity.

3. Need to add more diseases in perspective to oxidation and its mechanism. 

 4. Need to describe more Anthraquinone compounds 

Reviewer 2 Report

Comments and Suggestions for Authors

The results of the manuscript entitled “A Review on Bioactive Anthraquinone and Derivatives as the 2 Regulators for ROS” and authored by Zhao and Zheng showed that anthraquinones are of great significance for the development of natural antioxidants. Authors attributed the free radical scavenging activity of anthraquinone derivatives as the reason they are used as neuroprotective, hepatoprotective, anti-inflammatory and anti-cancer remedies. Because of their low solubility and rapid elimination rate, the oral absorption utilization ratio of anthraquinone is low. Therefore, authors concluded that it is necessary to modify the structure, add absorption promoters, or change drug dosage forms to improve the bioavailability. Before converging on anthraquinone, more in-depth introduction of natural products and biomolecules and their roles in promoting human health should be added. The following studies display biomolecules that promote general human health via multiple pathways: https://doi.org/10.1186/s41936-020-00177-9, PMID: 37450997, PMCID: PMC8007640. Patents of natural products (biomolecules) with reported effects analogous to the ones reported here should also be discussed. What time range of publication did this review article cover, what keywords did the search for literature include, what were the inclusion criteria, how many studies did the search find and how many were primary research vs review articles, of those, how many were selected for evaluation in this study,  and finally what criteria were used for selecting the articles that were reviewed (was it the subject of the study, its novelty or both), all that should be addressed.

Detailed comments

·       Proofreading is needed.

·       Abbreviations should be reviewed.

·       Maybe structuring the manuscript, a little different could ease up the flow of otherwise quite interesting concepts. Perhaps, discussing relevant literature based on anthraquinone’s applications rather than methodologies would help.

Comments on the Quality of English Language

Should the raised points be fully addressed, the revised version could then be re-evaluated for any possible future processing.

Reviewer 3 Report

Comments and Suggestions for Authors

A manuscript entitled “A Review on Bioactive Anthraquinone and Derivatives as the 2 Regulators for ROS” by Zhao et Zheng focuses on ROS-regulating effects of anthraquinones.

Major issues:

1.       ROS are of Janus-faced nature in cells. On one hand, they provide damage to macromolecules having detrimental effects. On the other hand, ROS signaling is of vital importance maintaining crucial cellular functions. The authors present  ROS contribution in a one-sided manner emphasizing exclusively their harmful impact.

2.       The authors should provide a brief summary of the intracellular antioxidant system (both enzymatic and non-enzymatic links), since its components are affected by antraquinones, which is discussed in the current review. These issues are partly discussed throughout the manuscript, but are not grouped, which makes the manuscript repetitive.

3.       In the Introduction section, ROS contribution to disease and ROS-targeting treatment strategies should be described in more detail.

4.       What is cancer prevention induced by anthraquinones. Evidence is not provided.

5.       The authors describe often contradictory effects of anthraquinones with no in-depth discussion and generalization (e.g., 4.1.5). The compounds described can be pro-apoptotic and anti-apoptotic. Possible explanation for this inconsistency should be provided.

6.       Subchapters mainly include description of the related studies with no critical analysis and generalization.

Minor issues:

1.       Subheading 4.1.5. Study on cellular experiments of natural anthraquinone compounds should be better replaced with Natural anthraquinone compounds in cell culture-based experiments

2.       Table 1 and Table 2. List of abbreviations should be provided.

3.       Page 8 line 252. MDA is a product of lipid peroxidation, so the phrase “mRNA expression of MDA” has no sense. Likewise, LDH was used probably as a cytotoxicity marker. Its release was analyzed, not expression.

4.       Page 8 line 261-263. This sentence is not clear. What is improvement of BAX and MAPKs?

Comments on the Quality of English Language

In general, the language is fine

Round 2

Reviewer 1 Report

Comments and Suggestions for Authors

The author has revised the manuscript as suggested. It can be accepted in the present form.

Reviewer 2 Report

Comments and Suggestions for Authors

none

Reviewer 3 Report

Comments and Suggestions for Authors

The authors have addressed the comments. 

Comments on the Quality of English Language

Some typos are still present in the manuscript.